# OpenReview forum: "Learning to Steer: Input-dependent Steering for Multimodal LLMs"
_NeurIPS.cc/2025/Conference — NeurIPS 2025 poster_

### Official Review · Reviewer_A2XA · 2025-07-01

**Clarity:** 3
**Significance:** 3
**Originality:** 3
**Rating:** 5
**Confidence:** 2

**Summary:**

This paper explores the challenges of steering behavior in multimodal large language models (MLLMs), focusing on input-specific steering to improve model outputs such as safety enforcement and hallucination mitigation. Traditional methods for steering rely on static steering vectors, which apply the same shift regardless of the input. The authors argue that such approaches are ineffective for cases where the desired behavior must vary based on the context, such as when a safe response is required for illegal or harmful queries. To address this limitation, they propose a method called Learn-to-Steer (L2S), which uses an auxiliary neural network to predict input-dependent steering vectors during inference.

**Questions:**

I have no additional questions.

**Ethical Concerns:**

["NO or VERY MINOR ethics concerns only"]

**Limitations:**

yes

**Quality:**

3

**Strengths And Weaknesses:**

The quality of the work is solid in terms of its experimental validation and theoretical contributions. The paper offers a practical solution to a well-established problem—safety and hallucination mitigation in MLLMs. The proposed L2S method is a clever extension of prior work. The authors provide a clear explanation of the approach, and their experiments show that L2S outperforms traditional methods, such as mean steering, across multiple tasks. The experimental results, especially in safety enforcement and hallucination mitigation, demonstrate a clear improvement over baseline methods. The paper also adequately addresses computational efficiency, highlighting that L2S maintains minimal overhead compared to other methods, making it practical for real-world applications.
The paper is well-written and clearly structured.
While steering methods have been explored before in the context of language models, the focus on input-specific steering vectors and the use of contrastive prompting to guide the model's behavior is novel. The authors' approach to fine-grained steering, which is tailored to specific inputs, is a significant departure from previous methods that applied a fixed steering vector to all inputs, and this represents an important conceptual and practical contribution.

In section 4.2, the authors use "safety" and "security" interchangeably. This should be changed to consistently say "safety".

To improve generalizability of findings the paper could evaluate different types of MLLMs or datasets.

---

> ### Author Rebuttal · Authors · 2025-07-30
>
> We thank the reviewer for recognizing the novelty of our approach, particularly the introduction of input-specific steering vectors learned via an auxiliary network — a key innovation that enables fine-grained, context-sensitive control. We also appreciate the acknowledgment of the method’s practicality and the importance of our contribution. We respond to their mentioned weaknesses below:
>
>
> **W1 (Writing edit):** Thanks for pointing out the error. We will make this change
>
>
> **W2 (Experiments on another MLLM):** Here we provide the results of our method using Qwen2-VL-7B-Instruct, for both applications of safety (on MMSafetyBench dataset) and hallucination (on POPE dataset). Our results remain largely consistent with results on LLaVA-v1.5 and favorable for L2S. For safety, L2S generalizes better compared to baselines in terms of simultaneously reducing harmfulness and achieving high expert-deference. The metrics and key takeaways are discussed below. The complete experimental details will be added in the paper.
>
>
> **Table: MMSafetyBench evaluation results.**
> These scores are reported using Qwen2-VL-7B-Instruct. Best result is in **bold**, and second best is in *italics*.
>
>
> | Metric | No-steering | Norm-Rnd | Mean-S | Mean-S (BA) | P2S* | L2S |
> | -------- |:----------:| :----------:| :--------------:| :--------:| :-----------:|:----------:|
> | $\mathbb{E}_{p \geq 0.5}[\text{Unsafe-score}(p)]$ ($\downarrow$) | 0.287 | 0.204 | 0.093 | **0.046** | 0.047 | *0.058* |
> | $\mathbb{E}_{p \geq 0.9}[\text{Unsafe-score}(p)]$ ($\downarrow$) | 0.193 | 0.107 | 0.050 | **0.024** | 0.019 | *0.032* |
> | ED-Score ($\uparrow$) | 0.184 | 0.013 | *0.408* | 0.316 | 0.565 | **0.592** |
> | Response-Quality ($\uparrow$) | 7.23 | **7.91** | 7.61 | 7.62 | 7.95 | *7.78* |
>
> The two most important takeaways for safety experiments remain the same:
>
>  1. With similar (and high) response qualities **L2S** is the only evaluated system that **performs strongly for both harmfulness reduction** (bridging gap to P2S in this case) **and expert deference** (best performing).
>  2. Since Mean-S mixes steering vectors from different prompt completions (lines 248-250) it performs worse than Mean-S(BA) in terms of reducing harmfulness but better at expert deference.
>
>
> **Table: POPE hallucination evaluation results.**
> The scores are reported per subset of POPE for Qwen2-VL-7B-Instruct. Best values are in **bold**.
>
> | Subset         | Metric       | No-steering | Mean-S   | **L2S**   | P2S*    |
> |----------------|--------------|:-----------:|:--------:|:--------:|:-------:|
> | **Random**     | Accuracy ↑   | 0.9175      | 0.9160   | **0.9300** | 0.9502  |
> |                | F1 score ↑   | 0.9570      | 0.9561   | **0.9637** | 0.9744  |
> | **Popular**    | Accuracy ↑   | 0.8926      | 0.8818   | **0.8958** | 0.9129  |
> |                | F1 score ↑   | 0.9433      | 0.9371   | **0.9450** | 0.9544  |
> | **Adversarial**| Accuracy ↑   | 0.8786      | 0.8709   | **0.8802** | 0.9144  |
> |                | F1 score ↑   | 0.9354      | 0.9310   | **0.9363** | 0.9553  |
>
>
> For Qwen, the original model is less prone to hallucination and performs better as it is also trained to be visually more grounded; hence, the improvement observed is still noticeable but less significant than in the case of LLaVA-1.5-7B.
> Nonetheless, both experiments clearly highlight advantages of L2S over mean-steering approaches for Qwen. We’ll be happy to add these in the main paper.
>
>
> **W2 (Experiments on another dataset):** We also conducted an experiment validating the safety steering for LLaVA-v1.5 on the **VLGuard** dataset [1]. Similar to results on MMSafetyBench, we obtain consistent results, wherein L2S generalizes simultaneously (and better compared to mean steering) to the two instantiations of safety behavior defined for VLGuard. The details are given below:
>
> - The dataset contains both unsafe input queries (unsafe images/instruction) and safe input queries. We define the instantiations of safety steering behavior as:
>   - Steer for safety/harmfulness feature for any unsafe query,
>   - No steering for any safe query. For the no-steering instantiation, the target steering vector is thus 0 vector.
>
> - We test the no-steering, mean-steering (Mean-S is same as Mean-S(BA)), and L2S methods. For unsafe test queries, we perform the harmfulness and response quality evaluation. For safe test queries, we evaluate (i) The Gemini win-rate between L2S and Mean-S responses, and (ii) Fraction of responses where steered output is exactly the same as the default unsteered model response. Precise details of all aspects will be added in the appendix.
>
> | Metric | No-steering | Mean-S(BA) | L2S |
> | -------- |:----------:| :----------:| :--------------:|
> | $\mathbb{E}_{p \geq 0.5}[\text{Unsafe-score}(p)]$ ($\downarrow$) | 0.0298 | 0.0140 | **0.0137** |
> | $\mathbb{E}_{p \geq 0.9}[\text{Unsafe-score}(p)]$ ($\downarrow$) | 0.0104 | 0.0058 | **0.0046** |
> | Response-Quality | **7.61** | 6.35 | 6.28 |
> | Gemini win-rate (Safe examples) (%) ($\uparrow$)  | XX | 28.2 | **71.8** |
> | Default answers (Safe examples) (%) ($\uparrow$) | 100 | 5.2 | **87.4** |
>
> **The results are consistent with previous results and favourable for L2S.** For similar response quality of steered responses, L2S obtains a slightly better unsafe score. However, most importantly, for more than 87% of the safe queries, the model output remains exactly the same, compared to 5% for mean-steering. Consequently, since L2S does not disturb the internal representations for safe queries, L2S responses are also consistently preferred over Mean-S responses for these queries.
>
> [1] Zong et al. “Safety Fine-Tuning at (Almost) No Cost: A Baseline for Vision Large Language Models”, ICML’24

---

### Official Review · Reviewer_Cgf1 · 2025-07-02

**Clarity:** 4
**Significance:** 4
**Originality:** 3
**Rating:** 5
**Confidence:** 4

**Summary:**

This paper proposes a lightweight auxiliary network based steering technique for Multimodal LLM's. Steering of Multimodal models is still largely unexplored making this an interesting new direction to be explored. The authors motivate their work well and substantiate the part of their proposed technique thoroughly. Their experimental results show a clear advancement over the state of the art. They also identify promising avenues for future research.

**Questions:**

1. Is the use of an auxiliary network truly novel? Please clarify.
2. What would happen if you had temporal signals such as video and audio in your framework?

**Ethical Concerns:**

["NO or VERY MINOR ethics concerns only"]

**Final Justification:**

I am satisfied by the authors' response in their rebuttal. They address both my novelty concerns and extension to video and audio crisply and convincingly with references and precise elaborations. I also appreciate the authors' restraint in not making overly strong claims on video and audio. I am therefore inclined to stay with my original rating of accept.
I have also gone through the other reviews and found that the other reviewers are positive and have provided constructive comments. I am pleased with the authors' constructive approach to criticism with their thorough and helpful response to the reviewer comments. I would recommend that the authors modify their paper with their response to reviewers in mind to make it stronger.

**Limitations:**

The authors discuss limitations by laying out avenues for future work which is a nice way to do it. I am satisfied with their discussion of limitations. The social impact of their work is positive by definition because steering techniques have all been developed with a view to increasing safety and accuracy.

**Paper Formatting Concerns:**

No problems with formatting

**Quality:**

3

**Strengths And Weaknesses:**

Strengths
1. Excellent literature review (Quality, Clarity)
2. Good motivation of proposed steering technique (Quality, Significance, Originality)
3. Significant improvement over the state of the art (Quality, Significance)
4. Well written paper (Quality, Clarity)

Weaknesses
1. I am not sure that the proposed technique is a significantly novel technique compared to the techniques it describes. Is the use of an auxiliary network that novel? The authors should clarify. Their technique is certainly sound and interesting.

---

> ### Author Rebuttal · Authors · 2025-07-30
>
> We thank the reviewer for their positive review and their praise regarding our motivation and the future research avenues it opens for future methods to take into account the input context during steering. We address your concerns/questions individually below:
>
> **W1/Q1 (Clarifying novelty of the approach):** Our proposed methodology has two main sources of novelty (Sec 1, line 53-56).
>
> - The first is the proposition of input-dependent steering, that is, steering in different directions depending upon the input context as the steering behavior can instantiate differently. This is a limitation with previous methods which do not modify the direction of steering and we first propose P2S, that uses input dependent contrastive prompts to address this limitation.
>
> - The second major novelty and **specific to the auxiliary network** is in realizing P2S in practice through L2S. P2S requires knowing the instantiation of behavior in advance to extract the steering vector which is not available during testing. L2S trains and employs a small auxiliary network $g$ to predict the steering vector from the input context (last token representation in our case). To the best of our knowledge, **the task of modeling input dependent steering vector through an auxiliary network is novel**. The architecture / training procedure of $g$ in itself is straightforward but the task for which it is trained is novel.
>
> We can make the first two contributions in Sec. 1, line 53-56 more clear to reflect these points.
>
>
> **Q2 (Steering for Audio/Video MLLMs):** This is an interesting question and a worthwhile direction to explore in the future. There are two key thoughts we have regarding these at the moment:
> - Multiple Audio based MLLMs (eg. SALMONN [1], Audio-Flamingo [2] models) have been designed similar to vision based MLLMs where the input signal is compressed to a set of tokens and given at once to the LLM part, either as prepended input tokens or through cross-attention. Even some SOTA Video MLLMs such as pLLaVA [3], or those based on processing video through an image patchwork like IG-VLM [4] adhere to this setup. In such cases, we would expect the methodology to carry over. In particular, we would expect the autoregressive design to take into account the temporal dimension and compress the information about the input signal in the context vector of last/final input tokens.
> - However, it should be mentioned that at the moment we are not aware of any Audio MLLM or Video MLLM based steering methods. In principle there is no particular reason why the steering methods should not be able to generalize, but it remains to be validated.
>
> In summary, while we remain hopeful that steering methods (including L2S) generalize to audio/video MLLMs, this remains to be validated and is an exciting future direction.
>
> [1] Tang et al. "SALMONN: Towards Generic Hearing Abilities for Large Language Models", ICLR 2024
>
> [2] Ghosh et al. "Audio Flamingo 2: An Audio-Language Model with Long-Audio Understanding and Expert Reasoning Abilities". ICML 2025
>
> [3] Xu et al. "PLLaVA : Parameter-free LLaVA Extension from Images to Videos for Video Dense Captioning", arXiV 2024
>
> [4] Kim et al. "An Image Grid Can Be Worth a Video: Zero-shot Video Question Answering Using a VLM". arXiV 2024

---

> > ### Comment · Reviewer_Cgf1 · 2025-08-03
> > **Response to rebuttal**
> >
> > I am satisfied by the authors' response in their rebuttal. They address both my novelty concerns and extension to video and audio crisply and convincingly with references and precise elaborations. I also appreciate the authors' restraint in not making overly strong claims on video and audio. I am therefore inclined to stay with my original rating of accept.

---

### Official Review · Reviewer_Le5q · 2025-07-02

**Clarity:** 4
**Significance:** 3
**Originality:** 3
**Rating:** 5
**Confidence:** 4

**Summary:**

This paper proposes a fine-grained steering approach to guide MLLMs towards enforcing specific behavior for safety enforcement
or hallucination mitigation. The proposed steering is an input-dependent linear shift. It trains a small network to predict a steering vector for an input. The experiments were conducted using safety enforcement and hallucination mitigation datasets and showed that the proposal outperformed existing steering methods that do not adapt to individual input.

**Questions:**

1.	The detailed information on contrast prompts would be helpful.
2.	Failure cases in the appendix are due to what?

**Ethical Concerns:**

["NO or VERY MINOR ethics concerns only"]

**Final Justification:**

I am satisfied with the reviewers' feedback. I confirmed the appendix, which is informative. I keep the rating.

**Limitations:**

yes.

**Paper Formatting Concerns:**

No.

**Quality:**

3

**Strengths And Weaknesses:**

Strength：
1.	The paper proposes a fine-grained steering for guidance of MLLMs. Existing steering methods of MLLMs only use a single steering vector for all inputs. The proposed fine-grained steering vector changes based on every input.

2.	It proposes a small network to predict the steering vector. The network is trained in advance. Steering vector prediction is quite efficient.

3.	It did two experiments – one used a safety enforcement dataset, and the other hallucination mitigation. The results show the proposal outperforms the prior works.

4.	The paper is well-written.


Weakness:
1.	Training the small network g is not clear.

2.	The detailed information on contrast prompts would be helpful.

3.	Failure cases are due to what?

---

> ### Author Rebuttal · Authors · 2025-07-30
>
> We thank the reviewer for their positive review and appreciation regarding the distinctiveness of our method by proposing to model the steering direction depending on the input.  We are also grateful for their positive comments about our experiments and writing clarity. We respond to their highlighted weaknesses/questions below:
>
> **W1 (Details for training $g$):** We provide details for training $g$ in **main text (Sec 3.3, Sec 4)** and **Appendix B.1**.
> To summarize:
>
> - We model $g$ as a 2-layer MLP with a bottleneck of size 100 and Tanh activation (Appendix B.1).
> - $g$ is trained to reconstruct the steering vector $z_{X, L^*}$ from the input context representation $h_{X, L'}$ using a reconstruction objective (Eq. 8 in Sec 3.3 and Appendix B.1).
> - It is optimized using Adam for 100 epochs with batch size 64 and a cosine learning rate scheduler (Section 4)
> - We also initialize the decoder weights using basis vectors obtained via dictionary learning (Semi-NMF/SVD) on the training steering vectors for stable training (Appendix B.1).
> - This process is very efficient as there is no need to back propagate through the base model $f$, and training completes in **under a minute** on standard hardware (Appendix B.1).
>
>   At inference (Eq. 9), $g$ predicts a steering vector based on the input context, which is then used to shift representations at generation time.
>
>
> **W2/Q1 (Contrastive prompt details):** We provide details for **contrastive prompts for safety experiments in Figure 1** , illustrating how we simulate different safety responses using positive and negative completions. Specifically (lines 212–215):
> - For the first 9 safety scenarios about harmful/illegal activities, we use a **common set of completions** simulating the input query being safe/harmful.
> - For the remaining 3 (legal, medical, financial), we use templates simulating input query requiring deferring to experts for appropriate handling.
>
> For the **hallucination task** (lines 279–280), the contrastive prompts are constructed based on **binary object presence questions** (e.g., “Is there a watch in the image?”).
> - A **positive prompt** consists of the model giving the **correct answer** (e.g., "No, there is no watch.") and continuing its response naturally.
> - A **negative prompt** consists of the model giving the **incorrect (hallucinated) answer** (e.g., "Yes, the person is wearing a watch.") and continuing in the same style.
>
> This setup enables the model to contrast grounded vs. ungrounded generations.
>
> We have also provided concrete examples of these completions in Appendix B.4 (Steering details) for clarity. We will update the paper to make this organization more clear and explicitly point readers to Figure 1 and Appendix B.4.
>
>
>
> **W3/Q2 (Failure cases):** Many failure cases are likely because the steering strength is not strong enough for certain inputs. For example, in Fig. 6 (Right) Appendix, the output is steered towards giving a safe response (green text at the end, asking to avoid engaging in illegal activities). However, it's not strong enough as the model still outputs certain harmful instructions at the start (red text). This hypothesis is also supported by the fact that we achieve near perfect harmfulness reduction for high $\alpha$, albeit at the cost of response quality (Fig. 10 (Left) Appendix). Another source of error as in Fig. 6 (Left) Appendix, which is more rare, is that the steering direction may introduce some unwanted features (such as smoking in this case). This could likely be reduced by improving the steering vector extraction process, for example with more optimal selection of contrastive prompts.
>
> As already noted in appendix A.1, it is worth highlighting that we have observed such failure cases with other steering baselines too. Nevertheless, an interesting direction for future work that we can add to the dedicated section would be to try and predict the steering vector direction and norm separately for each input or integrate information about appropriate norm during P2S extraction. We will add this discussion to appendix A.1 where we showcase these figures.

---

> ### Comment · Reviewer_Le5q · 2025-08-05
>
> I did not notice the appendix in the first review. It is informative. I was positive and keep the rating.

---

### Official Review · Reviewer_bck9 · 2025-07-03

**Clarity:** 3
**Significance:** 3
**Originality:** 3
**Rating:** 4
**Confidence:** 3

**Summary:**

This paper introduces L2S, an innovative input-dependent steering method for MLLMs that addresses critical issues of hallucination and safety. Moving beyond static steering, L2S leverages a small auxiliary network to predict input-specific steering vectors, enabling dynamic adjustments to MLLM behavior based on individual queries. This approach significantly enhances model safety and reduces undesirable outputs with minimal computational overhead, offering a highly efficient and effective solution for reliable MLLM deployment.

**Questions:**

1、The core method L2S relies on ground-truth steering vectors generated by P2S. However, manually crafting high-quality contrastive prompts for each new task or behavior can be time-consuming and may require domain expertise. Can this approach realistically scale to novel or previously undefined behaviors?
2、The authors should include an ablation study on L to better isolate and evaluate its effect.
3、The authors adopt a lightweight 2-layer MLP as the auxiliary network, which is a reasonable design choice. However, is its representational capacity sufficient? Did the authors consider more complex architectures to potentially improve performance?

**Ethical Concerns:**

["NO or VERY MINOR ethics concerns only"]

**Limitations:**

yes

**Quality:**

3

**Strengths And Weaknesses:**

Strengths:
1、The proposed L2S method directly tackles two paramount challenges in MLLMs: hallucination and safety.
2、The design, particularly the use of a lightweight network to predict steering vectors, makes L2S computationally very efficient.

Weaknesses:
1、The evaluation is limited to LLaVA1.5, without sufficient coverage of other representative MLLM architectures like Qwen-VL.
2、L2S operates by applying a linear shift in the latent space. However, the paper does not delve deeply into how this shift specifically alters the MLLM's internal states or its understanding of input concepts.
3、The paper focuses on mitigating hallucination and enforcing safety. However, guiding the model towards these specific behaviors might affect or degrade its performance on other important dimensions such as informativeness. The authors should conduct experiments to evaluate L2S's impact on other desired qualities of MLLM responses.

---

> ### Author Rebuttal · Authors · 2025-07-30
>
> We are grateful to the reviewer for the positive review and for recognizing the novelty of our work in addressing the critical issues of hallucination and safety in MLLMs via advancement towards input-dependent steering. We respond to their concerns pointwise below:
>
> ## Weaknesses
>
> **W1 (Experiments on other MLLMs):** Here we provide the results of our method using Qwen2-VL-7B-Instruct, for both applications of safety (on MMSafetyBench dataset) and hallucination (on POPE dataset). Our results remain largely consistent with results on LLaVA-v1.5 and favorable for L2S. For safety, L2S generalizes better compared to baselines in terms of simultaneously reducing harmfulness and achieving high expert-deference. The metrics and key takeaways are discussed below. The complete experimental details will be added in the paper.
>
>
> **Table: MMSafetyBench evaluation results.**
> These scores are reported using Qwen2-VL-7B-Instruct. Best result is in **bold**, and second best is in *italics*.
>
>
> | Metric | No-steering | Norm-Rnd | Mean-S | Mean-S (BA) | P2S* | L2S |
> | -------- |:----------:| :----------:| :--------------:| :--------:| :-----------:|:----------:|
> | $\mathbb{E}_{p \geq 0.5}[\text{Unsafe-score}(p)]$ ($\downarrow$) | 0.287 | 0.204 | 0.093 | **0.046** | 0.047 | *0.058* |
> | $\mathbb{E}_{p \geq 0.9}[\text{Unsafe-score}(p)]$ ($\downarrow$) | 0.193 | 0.107 | 0.050 | **0.024** | 0.019 | *0.032* |
> | ED-Score ($\uparrow$) | 0.184 | 0.013 | *0.408* | 0.316 | 0.565 | **0.592** |
> | Response-Quality ($\uparrow$) | 7.23 | **7.91** | 7.61 | 7.62 | 7.95 | *7.78* |
>
> The two most important takeaways for safety experiments remain the same:
>
>  1. With similar (and high) response qualities **L2S** is the only evaluated system that **performs strongly for both harmfulness reduction** (bridging gap to P2S in this case) **and expert deference** (best performing).
>  2. Since Mean-S mixes steering vectors from different prompt completions (lines 248-250) it performs worse than Mean-S(BA) in terms of reducing harmfulness but better at expert deference.
>
>
> **Table: POPE hallucination evaluation results.**
> The scores are reported per subset of POPE for Qwen2-VL-7B-Instruct. Best values are in **bold**.
>
> | Subset         | Metric       | No-steering | Mean-S   | **L2S**   | P2S*    |
> |----------------|--------------|:-----------:|:--------:|:--------:|:-------:|
> | **Random**     | Accuracy ↑   | 0.9175      | 0.9160   | **0.9300** | 0.9502  |
> |                | F1 score ↑   | 0.9570      | 0.9561   | **0.9637** | 0.9744  |
> | **Popular**    | Accuracy ↑   | 0.8926      | 0.8818   | **0.8958** | 0.9129  |
> |                | F1 score ↑   | 0.9433      | 0.9371   | **0.9450** | 0.9544  |
> | **Adversarial**| Accuracy ↑   | 0.8786      | 0.8709   | **0.8802** | 0.9144  |
> |                | F1 score ↑   | 0.9354      | 0.9310   | **0.9363** | 0.9553  |
>
>
> For Qwen, the original model is less prone to hallucination and performs better as it is also trained to be visually more grounded; hence, the improvement observed is still noticeable but less significant than in the case of LLaVA-1.5-7B.
> Nonetheless, both experiments clearly highlight advantages of L2S over mean-steering approaches for Qwen. We’ll be happy to add these in the main paper.
>
>
>
> **W2 (Understanding effect of steering vectors)**: We agree that there are deeper connections to explainability that can be explored further. While it wasn't the focus of our work, we do provide primary analysis of the steering vectors. In **Figure 9** (appendix), we show:
>
> - **Fig 9 (Left)**: Average pairwise cosine similarities between P2S steering vectors for different sets of contrastive prompts. **High intra-behavior similarity** and **low inter-behavior similarity** indicate that different behavior instantiations are encoded in largely distinct directions.
>
> - **Fig 9 (Right)**: A **t-SNE visualization** of steering vectors generated using the same contrastive prompt across inputs. Even when using the same prompt completions, the vectors still cluster by input scenario, showing that they capture some **context information**.
>
> A possible direction for future work could be to train the L2S model with additional constraints (e.g., **sparsity of intermediate activations**), which may lead to a more interpretable second layer acting as a **dictionary of steering directions**.
>
>
> **W3 (Evaluating Quality of MLLM responses)**: We completely agree that during steering, it is essential to ensure that the output quality is not degraded on other crucial aspects.
>
> - In this regard, we would like to emphasize that the **Response quality evaluation** in Tab. 1, 3 (descriptions in line 236-242, line 290-294) assessed this aspect for all our experiments, designed individually for safety and hallucination experiments. For safety experiments, the evaluation quantifies if the steered response remains **coherent** and **relevant to the input context**. For hallucination experiments, we quantify how the **steered responses rate** against the unsteered responses by the win-rate metric. Complete details about the evaluations can be found in Appendix B.3.
>
> - We also qualitatively assessed that the steered responses remained relevant to the input query and coherent.
>
> - Furthermore, through our experiments on VLGuard (last point to Reviewer A2XA), we also demonstrate that L2S can be trained to keep the original response unaffected (i.e. not steer) depending upon the input context.
>
> - If needed, we can also add perplexity analysis of the steering methods, wherein we observed that all the steering baselines showcased low perplexity (relative to vocabulary size), indicating the model remains highly confident even after steering.
>
>
> That said, we agree that quantifying all desirable traits (e.g., informativeness, helpfulness, engagement) is challenging, especially in multimodal settings. We will update our limitations. This has also remained a limitation for previous steering methods (main paper refs [18, 24, 56]). While we make some progress in this regard with the response quality evaluation, there remains space for further improvement in the future.
>
>
> ## Questions
>
> **Q1 (Scaling contrastive prompting to novel behaviors)**: This is a fair point. The first limitation we highlight in lines 317-320 is along very similar theme. Namely, P2S significantly speeds up the process of finding some operational prompt pair, as it allows for sample-wise testing of contrastive prompts. However, indeed it is not easy to find the optimal contrastive prompts as the possible number of completions is extremely large.
>
> There are no prior guarantees one can have regarding any novel behavior. Nevertheless, the success of previous contrastive methods for steering LLMs for different types of tasks (main paper refs [38, 50]) positively reflects on our approach and contrastive steering methods in general, to scale to novel behaviors. Moreover, we believe that for a given contrastive prompt and novel behavior, if there is a fast and reliable way to automatically evaluate the steering effectiveness, there can be possibilities to automate the generation of suitable P2S contrastive prompts, through another LLM. This can be a very interesting future direction to pursue.
>
>
>
> **Q2 (Analysis of L):** Analysis for both L*, L’ is available in Appendix B.2 (Figures 10, 11, 12, 13) for both safety and hallucination tasks.
>
>
> **Q3 (Complexifying architecture of $g$):**
> This is an interesting question. The representation capacity was sufficient for our experiments.
>
> - We validated that through the cosine similarity/reconstruction error of L2S predicted vectors with P2S extracted vectors, which has a high absolute value and also noticeably higher than mean-steering vector (Fig. 11, Appendix). For context layer $L’=30$, cosine similarity for L2S was around 0.95-0.96, while the mean-steering vector similarity was around 0.73. We thus believed the prediction quality was sufficiently high with a compact 2-layer MLP $g$.
>
> - Moreover, even before training any $g$, we analyzed the structure of the input space and output space of L2S model for our experiments. In particular, (i) the steering vectors for a single instantiation of a behavior tend to be similar to each other (Fig. 9, Left), (ii) The P2S steering vectors themselves, even when extracted with a fixed prompt are clustered according to the input context (Fig. 9, Right). This led us to believe that the task of predicting input specific steering vectors from input context should not require a highly complex prediction network.
>
> Nevertheless, for a complex setting, it is possible that simply using the last token as context is not enough, in which case one could consider expanding the context and/or complexifying the architecture of L2S. Our framework should easily account for this possibility.

---

### Decision · Program_Chairs · 2025-09-17

**Decision:**

Accept (poster)

**Comment:**

# Summary
The paper proposes training a lightweight auxiliary network to produce input-specific steering vectors, which can be used to control the output behaviors of MLLMs. The method provides an efficient and adaptive approach for MLLM steering. Experiments on safety alignment and hallucination mitigation show the effectiveness of the proposed L2S method, which outperforms SOTA methods based on static steering vectors.

# Strengths
- Input-specific steering is novel and effective.
- The paper demonstrated that a small network suffices to learn how to generate steering vectors.
- The experiments showcase the advantage of the proposed method on two important tasks.
- The paper is well-written.

# Weaknesses
- The technical novelty is limited due to its similarity to CoCoOp ("Conditional Prompt Learning for Vision-Language Models" CVPR 2022), in which an auxiliary small network is trained to produce input-specific prompts for VLMs.
- The experiments are limited to one or two VLMs and two steering tasks.
- The paper can be improved by an in-depth analysis of how the L2S steering vectors change the model's internal states, dynamics, and behaviors.

# Reasons to Accept
- The MLLM steering problem is a popular topic in the community, and this paper provides novel insights into generating input-specific steering vectors.
- The proposed idea is intuitive and effective in the experiments.
- The paper presentation is clear and easy to understand.

# Discussion Summary
- In the rebuttal, the authors provided additional experiments on more MLLMs and tasks. They also provide further details and explanations of the proposed method.
- Overall, reviewers are satisfied with the rebuttal and additional results. All reviewers support the acceptance of this paper.